# Effect of Sugar Content on Quality Characteristics and Shelf-Life of Probiotic Dry-Fermented Sausages Produced by Free or Immobilized *Lactobacillus casei* ATCC 393

**DOI:** 10.3390/foods8060219

**Published:** 2019-06-21

**Authors:** Marianthi Sidira, Gregoria Mitropoulou, Alex Galanis, Maria Kanellaki, Yiannis Kourkoutas

**Affiliations:** 1Food Biotechnology Group, Section of Analytical Environmental and Applied Chemistry, Department of Chemistry, University of Patras, GR-26500 Patras, Greece; sidiramania@yahoo.gr; 2Laboratory of Applied Microbiology and Biotechnology, Department of Molecular Biology & Genetics, Democritus University of Thrace, 68100 Alexandroupolis, Greece; grigoriamitropoulou@gmail.com (G.M.); agalanis@mbg.duth.gr (A.G.)

**Keywords:** probiotics, meat products, wheat, multiplex PCR, fermentation, extended shelf-life

## Abstract

The aim of the present study was to investigate the effect of sugar content (0, 0.30, and 0.60%) on quality attributes and shelf-life of dry-fermented sausages stored for 66 days containing free or immobilized *Lactobacillus casei* ATCC 393 on wheat. For comparison, dry-fermented sausages with no starter culture were also produced. Physicochemical characteristics ranged within the levels usually observed in fermented sausages, while a drastic decrease was recorded in numbers of enterobacteria, staphylococci, and pseudomonads during ripening in all cases. Noticeably, sugar addition and the probiotic culture resulted in significant increase of shelf-life, whereas levels of *L. casei* ATCC 393 after 66 days of ripening persisted above 6 log cfu/g. Sugar addition had a positive effect on sensory attributes; although all products were of high quality, the immobilized cells provided a distinctive characteristic aroma and a fine taste.

## 1. Introduction

There is a growing interest in designing and developing novel foods containing beneficial microorganisms like bifidobacteria and lactic acid bacteria (LAB). Indeed, functional foods have a great prospect in human health maintenance [1]. Definitely, a probiotic-rich diet has been associated with the prevention and potential treatment of several severe digestive disorders [1,2]. Probiotic foods often target high cell densities (10^6^–10^7^) of health-promoting microbes, aiming at survival of sufficient number of cells at the gastrointestinal (GI) tract based on a reasonable amount of food consumed [3]. However, maintenance of high viability levels is not always fulfilled.

Incorporation of probiotic bacteria into a food matrix represents a fully new challenge for the industry because of the microbial interactions with food constituents, as well as due to the harsh conditions usually encountered during food processing and storage, which often result in severe reduction of viable cell counts [4,5]. To overcome such obstacles and maintain high viability, activity, and functionality, cell immobilization on food-grade natural supports is a promising alternative [6,7,8]. Cereals are expected to act as delivery vehicles of the microbial cultures to the human gut when used as immobilization supports because they are rich in prebiotic constituents, the functional properties of which should be further investigated [9,10]. Wheat is currently the principal staple food for nearly one-third of the world’s population, providing >50% of the total daily energy intake [11] and it contains significant amounts of non-digestible carbohydrates, representing a new class of nutrients with prebiotic action [12,13].

Dry-fermented sausages are typical Mediterranean meat-products, the acceptability of which is strongly influenced by their taste and flavour. During their production, sugars and/or skim milk powder are usually added to improve the fermentation rate and acid production [14,15,16,17]. Although sugar addition is a well-established practice in the meat industry to facilitate the fermentation process and ensure safety, determination of the optimum initial sugar concentration is always considered necessary during new product development [15,16,17,18], especially in additive-free products.

Despite the fact that a few previous research efforts for inserting functional cultures in dry-fermented sausages are available in the literature [16,19,20,21,22], survival of the probiotic strains in adequate amounts after the manufacturing process is not yet documented [21,22]. Thus, commercial fermented sausages containing probiotic microorganisms are not available yet.

*Lactobacillus casei* ATCC 393 has been previously incorporated into food products [7,19,20,23] to confer probiotic properties [2,24,25] due to its acid-tolerant nature and activity against food spoilage and food-borne pathogenic microorganisms. Importantly, cell immobilization resulted in significantly increased survival rates over free cells during heat-treatment of dry-fermented sausages [19], as the probiotic cells were detected above the minimum levels required for conferring a beneficial effect using strain-specific multiplex PCR [19]. Likewise, immobilized *L casei* ATCC 393 cells were proposed as a protective shield against spoilage of probiotic dry-fermented sausages [20]. However, the effect of the initial sugar concentration on the quality characteristics and the shelf-life of probiotic dry-fermented sausages containing free or immobilized *L. casei* ATCC 393 cells was not studied.

In the present study, data suggesting significant enhancement of microbial safety due to advanced fermentation rate, improvement of sensory attributes, and prolongation of shelf-life in probiotic dry-fermented sausages with higher sugar content are presented.

## 2. Materials and Methods

### 2.1. Bacterial Strain and Cell Immobilization

*Lactobacillus casei* ATCC 393 (DSMZ, Braunschweig, Germany) was grown at 37 °C for 72 h on MRS Broth (LabM, Keywood, UK). The wet cells obtained after centrifugation were used directly for cell immobilization or for production of probiotic dry-fermented sausages.

Cell immobilization was performed by introducing 500 g of boiled and sterilized wheat grains in a 2 L MRS Broth culture containing 8 g (wet weight) of *L. casei* cells [8]. The mixture was allowed to ferment at 37 °C for 48 h without agitation.

### 2.2. Production of Probiotic Dry-Fermented Sausages

Dry-fermented sausages were prepared and ripened according to traditional techniques, as described previously [19]. Briefly, a batch consisting of ground pork meat (2.0 kg), lard (0.5 kg), ground orange peel (25.0 g), ground leak (412.5 g), sodium chloride (50.0 g), pepper (3.75 g), red pepper (3.75 g), cumin (3.75 g), ground garlic (1.25 g), and oregano (10.0 g) was inoculated with the probiotic cultures and dry-fermented sausages containing 30 g of immobilized *L. casei*/kg of stuffing mixture (samples IM) or 1.0 g wet weight of free *L. casei*/kg of stuffing mixture (samples Fr) were prepared. For comparison reasons, sausages with no starter (samples NC) were also produced.

To investigate the effect of the initial sugar concentration, sucrose and lactose were added at the above stuffing mixture at various concentrations prior to sausages preparation. Thus, addition of ≈0.60% total sugar (0.50% sucrose and 0.10% lactose) or ≈0.30% total sugar (0.25% sucrose and 0.05% lactose) was tested. Products with no sugar were also produced and served as controls.

After mixing, the stuffing of natural casings produced fresh sausages. Ripening was carried out at room temperature (19–23 °C, RH 82–85%) for 11 days. Then, all products were packaged aerobically in sterilized plastic pouches sealed aseptically and stored in a refrigerator (4–6 °C, RH 72–75%) for up to 66 days.

All experiments were carried out in triplicate (3 independent batches were prepared) and duplicate samples from each treatment were collected daily for the first 11 days and then weekly and subjected to chemical, microbiological, and strain-specific multiplex PCR.

### 2.3. Physicochemical Analysis

Titratable acidity was determined as described previously [19,20], while pH was determined with a pHmeter (WTW pH-330i pHmeter, Xylem Analytics, Weilheim, Germany). Water activity (Aw) determination was carried out with a calibrated electric hygrometer (HygroLab, Rotronic, Switzerland) according to the manufacturer’s instructions. Weight loss was calculated by weighing the sausages just after stuffing (day 1) until 11 days of ripening and by reweighing on the 2nd, 3rd, 4th, 9th, and 11th day. The differences in weight were expressed as % percentage of the initial weight.

### 2.4. Microbiological Analysis

Microbiological analysis was performed weekly, as described previously [19,20]. Briefly, representative 10 g portions of duplicate sausages samples from the interior were blended with 90 mL of sterilized ¼ Ringer’s solution and subjected to serial dilutions.

The following microbial species were determined: (i) total aerobic counts on plate count agar (Fluka, St. Gallen, Switzerland) at 30 °C for 48 h, (ii) lactobacilli (Gram (+), catalase (-)) on acidified MRS agar (Fluka) at 37 °C for 48 h anaerobically (Anaerobic jar, Anerocult C, Merk, Germany), (iii) enterobacteria on violet red bile glucose agar (Fluka) at 37 °C for 24 h, (iv) staphylococci on Baird Parker egg yolk tellurite medium (Fluka) at 37 °C for 48 h and confirmed by a positive coagulase test, (v) pseudomonads on pseudomonas CFC selective agar at 25 °C for 72 h, (vi) yeasts and moulds on malt agar (Fluka) (pH was adjusted to 4.5 by sterile solution of 10% lactic acid) at 30 °C for 48 h, and (vii) clostridia on TSC Agar at 37 °C for 24 h anaerobically (Anaerobic jar, Anerocult C, Merk). All incubations were further extended up to 120 h, but no extra colonies were observed. Gram staining and catalase tests were performed for lactic acid bacteria confirmation. Results are presented as log of mean colony-forming units on solid media culture plates containing between 30 and 300 colonies per gram of dry-fermented sausage.

### 2.5. Determination of L. casei ATCC 393 Levels

To confirm the presence/absence of our strain in dry-fermented sausages, a previously described methodology was followed [19,26].

### 2.6. Preliminary Sensory Evaluation

The effect of the nature of the starter probiotic culture and the sugar addition on the sensory characteristics of the dry-fermented sausages was studied. The new products were also compared to a similar type commercial product. Raw slice samples of approximately 25 g were served in random order at room temperature. Sensory evaluation was conducted in duplicate by 11 laboratory members familiar with fermented sausages taste (specialists) using locally approved protocols. The panel was unaware of the identity of the samples and was asked to give scores on a 0–5 continuous scale (0: unacceptable, 5: exceptional) for attributes grouped into five categories: appearance, colour, aroma, consistency, and overall quality. No reference materials were used for sensory attributes and no training session was involved to standardize the attribute intensities. Panellists used water to clean their palates between samples.

Spoilage was determined macroscopically and by using sensory tests, as previously described [19].

### 2.7. Experimental Design and Statistical Analysis

All treatments were carried out in triplicate (3 independent batches of sausages were prepared). The experiments were designed and analyzed statistically by ANOVA. Specifically, 3-way ANOVA was applied for physicochemical parameters and microbiological analysis (sugar addition, the nature of the starter culture, and the ripening time were the 3 factors) and 2-way ANOVA in the preliminary sensory evaluation (sugar addition and the nature of the starter culture were the 2 factors). Duncan’s multiple range test was used to determine significant differences among results (coefficients, ANOVA tables, and significance (*p* < 0.05) were computed using Statistica v.10.0 (StatSoft, Tulsa, OK, USA).

## 3. Results

### 3.1. Physicochemical Analysis

The effect of sugar addition, the nature of the starter culture, and the ripening time on physicochemical characteristics were studied (Figure 1). All parameters, (water activity (Aw), weight loss, pH, and titratable acidity) were significantly (*p* < 0.05) affected by all factors. Significant (*p* < 0.05) interactions between all factors were observed in pH, while interactions (*p* < 0.05) between sugar addition and the nature of the starter culture, sugar addition and the ripening time, and the nature of the starter culture and the ripening time were noted in weight loss. Likewise, significant interactions (*p* < 0.05) between sugar addition and ripening time, the nature of the starter culture and the ripening time, as well as among all factors were recorded in titratable acidity.

A significant (*p* < 0.05) increase in titratable acidity and a significant (*p* < 0.05) decrease in pH, water activity (Aw), and weight loss were noted during maturation in all samples. However, an increase (*p* < 0.05) in pH was observed at the late stages of ripening, probably due to ammonia formation by amino acid degradation [27]. The highest (*p* < 0.05) titratable acidity (1.93% lactic acid) was observed in dry-fermented sausages with free cells containing 0.60% sugars after 45 days of ripening.

### 3.2. Shelf Life and Microbiological Analysis

Sugar addition and the probiotic culture resulted in significant (*p* < 0.05) increase of shelf-life (Table 1). Of note, spoilage was monitored up to 66 days.

The association of the microbial groups examined during ripening of the probiotic dry-fermented sausages is presented in Figure 2. The effect of sugar addition, the nature of the starter culture, and the ripening time on microbial counts was studied. Total aerobic counts were significantly (*p* < 0.05) affected only by the ripening time, while numbers of pseudomonads by both the nature of the starter culture and the ripening time (*p* < 0.05). In contrast, all factors had a significant (*p* < 0.05) effect on counts of lactobacilli, enterobacteria, staphylococci, and yeasts/moulds. Significant (*p* < 0.05) interactions between all factors were observed in counts of enterobacteria, staphylococci, and yeasts/moulds, whereas interactions (*p* < 0.05) between sugar addition and the ripening time, the nature of the starter culture and the ripening time, and among the three factors were noted in total aerobic counts. Similarly, interactions (*p* < 0.05) between sugar addition and the nature of the starter culture, the nature of the starter culture and the ripening time, and among all factors were observed in pseudomonads counts. Finally, significant interactions (*p* < 0.05) between the nature of the starter culture and the ripening time were recorded in lactobacilli counts.

Total aerobic counts and levels of lactobacilli and yeasts/moulds remained high during ripening. On the contrary, a drastic decrease (*p* < 0.05) was observed in enterobacteria, pseudomonads, and staphylococci counts (Figure 2). Noticeably, clostridia were not detected in any sample during the whole ripening period.

### 3.3. Determination of L. casei ATCC 393 Levels

To determine the levels of the used strain during ripening of the probiotic sausages, identification after cell counting on MRS agar plates was carried out by multiplex PCR assay using specific primers that generated two unique PCR products of 67 bp and 144 bp for *L. casei* ATCC 393 [26]. A set of universal lactobacilli primers generating a PCR product of 340 bp was used as positive control [28]. *L. casei* ATCC 393 was identified at levels ≥6 log cfu/g in all samples containing immobilized or free cells after 66 days of ripening (Figure 3). As expected, the above strain was not detected in sausages produced with no starter culture.

### 3.4. Preliminary Sensory Evaluation

Sugar addition affected significantly (*p* < 0.05) the appearance, while the nature of the starter culture had a significant (*p* < 0.05) effect on sausage aroma. Noticeably, the colour and overall acceptance were significantly (*p* < 0.05) affected by both factors. In contrast, no significant (*p* > 0.05) effect on product consistency was observed.

Generally, sugar addition improved the appearance and colour, while the dry-fermented sausages produced using immobilized cells had a distinctive, characteristic aroma, which lasted until the end of the ripening period (66 days). All products containing the probiotic culture were approved and accepted by the panel when compared to a similar type commercial product and the preliminary sensory test ascertained the fine taste and the overall high quality.

## 4. Discussion

Functional starter cultures suitable for dry-fermented sausage production are expected to meet health promoting, food safety, shelf-life, technological effectiveness, and economic feasibility criteria. Although the use of probiotic bacteria is already a common practice in dairy products, the food industry is looking for functional meat products with improved quality characteristics. Thus, the scope of the present study was to investigate the effect of sugar addition on quality parameters of probiotic dry-fermented sausages containing free or immobilized *L. casei* ATCC 393 on wheat grains, since sugars, as well as skim milk powder are usually added to facilitate the fermentation process and acid production [14,15,16,17]. The strategy adopted aimed at maintaining survival of the probiotic strain at adequate levels for providing the health effect at the time of consumption, ensuring the microbial safety and improving the sensory characteristics of the new products, and finally extending the products’ shelf life. In this context, two initial sugar concentrations were tested, and spoilage was assessed for a period higher than 2 months, considering that the dry-fermented sausages produced were additives-free (contained no nitrates and/or nitrites), according to the requirements of our industrial partner.

The physicochemical parameters of the probiotic dry-fermented sausages ranged in levels usually observed in dry-fermented sausages [27,29], except titratable acidity which was slightly increased. However, sensory quality deterioration, usually attributed to overacidity, was not obvious in our samples. Similar observations were also reported previously [19,20].

The significant increase of preservation time observed in sausages produced with the inclusion of probiotic cultures compared to sausages produced with no culture for longer time periods before the observation of sparse white spots (Table 1) supported our initial hypothesis for achieving an extension of the shelf-life. Monitoring for up to 66 days revealed that free or immobilized *L. casei* ATCC 393 exerted a protective effect against spoilage on sausages independently of the concentration of sugars tested, although protection tended to increase in high sugar concentrations. Similar results were previously observed [19,20] and could be attributed to the higher titratable acidity and lower pH and Aw recorded. Although high initial counts of enterobacteria, pseudomonads, and staphylococci were detected at day 1, probably due to poor sanitation conditions, they were significantly reduced during ripening. The domination of LAB and yeasts and moulds and the drastic decrease of enterobacteria, pseudomonads, and staphylococci counts has also been reported before [17,18,19,20]. Enterobacteria may originate by the animal tissues, contaminated during slaughter and animal quartering, as several members of this family are natural inhabitants of the GI tract. Staphylococci are poorly competitive in the presence of actively growing acidic bacteria, seldom overcoming one log cfu/g during ripening [22]. Although dry-fermented sausages are generally considered safe, survival and even growth of certain pathogens should not be excluded. In our study, counts of presumptive pathogenic species dropped to undetected levels during ripening, obviously due to combination parameters of recipe and process, according to the “multiple hurdles concept” [30]. Fungal surface colonization of ripening sausages begins with salt and acid-tolerant yeast species, which are considered essential for sausage flavour formation [31], although their role is not well characterized. Along with the decrease in Aw, a shift towards moulds usually occurs. However, their overgrowth may result in quality deterioration, spoilage, and mycotoxin production despite the fact that it is a desired and characteristic feature in many products in some countries. In dry-fermented sausages, a significant decrease of bacterial viability is often associated to the low water activity and pH [21]. In our study, the levels of both free and immobilized *L. casei* ATCC 393 ranged above the minimum required concentration for a probiotic effect (≥6 log cfu/g) after 66 days of ripening. The above results were in accordance to previously published studies by our group [19,20]. Likewise, reports on the survival of free probiotic cultures during ripening [32,33], as well as on the positive effect of microencapsulation on the improvement of cell survival during production of dry-fermented sausages are available in literature [34]. However, probiotic strain identification was not efficiently accomplished in these studies due to the lack of an accurate, reliable, and sensitive detection assay.

Production of functional foods containing viable probiotic strains at the minimum essential levels for providing a health effect until the time of consumption is a technological challenge. Wheat grains served as an immobilization support of *L. casei* cells in the present study, since they are considered a nutritious food constituent, rich in prebiotic dietary fibres [13]. Prebiotic supports may represent a useful vehicle for the transfer of probiotic cells in the GI tract. Although meat products are seldom perceived as “healthy foods”, the marketing potential of fermented sausages still requires enhanced scientific research effort, mainly oriented in the selection of suitable means for targeted probiotic cell delivery in the gut.

## 5. Conclusions

Our data suggested that sugar addition had a positive effect on quality characteristics of dry-fermented sausages and the inclusion of immobilized cells of *L. casei* ATCC 393 provided a distinctive characteristic aroma to the product. Importantly, both free and immobilized cells survived during the maturation process. However, more research is still required mainly towards to the selection of appropriate vehicles for targeted delivery of probiotic bacteria to various sites within the GI tract and able to ensure high cell viability in the food matrix up to the time of consumption.

## Figures and Tables

**Figure 1 foods-08-00219-f001:**
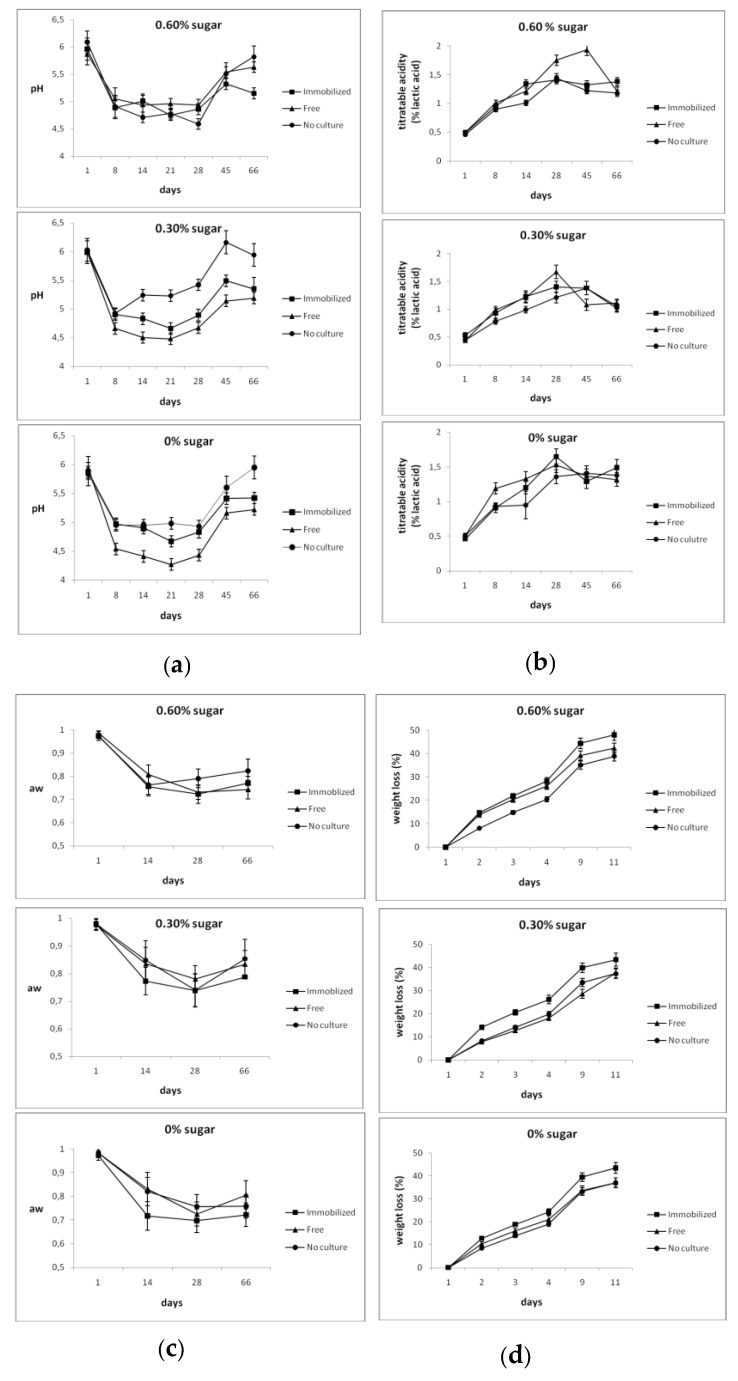
Effect of probiotic starter culture and sugar content on physicochemical parameters during ripening of probiotic dry-fermented sausages: (**a**) pH, (**b**) titratable acidity, (**c**) water activity (Aw) and (**d**) % weight loss. Error bars indicate standard deviations (*n* = 3).

**Figure 2 foods-08-00219-f002:**
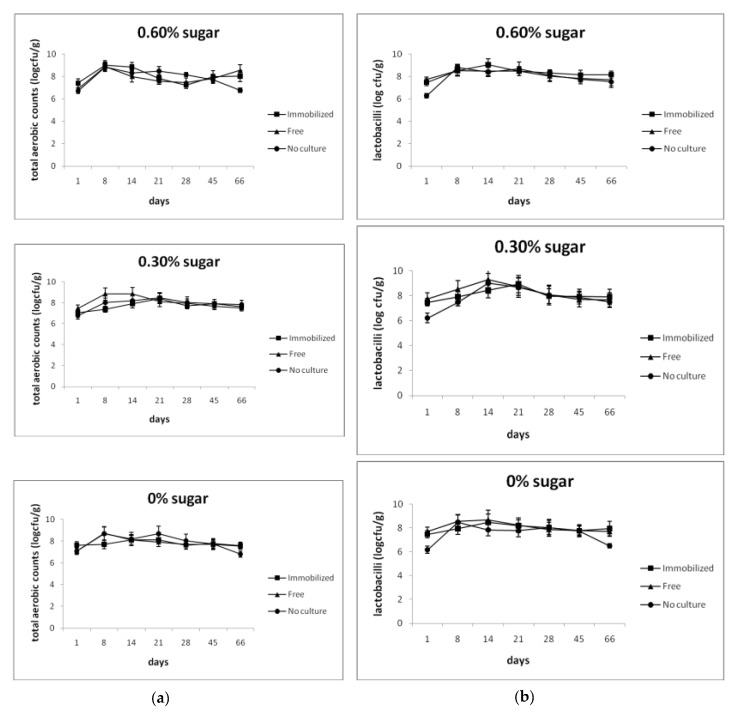
Effect of sugar addition and probiotic starter culture on microbial association during ripening of dry-fermented sausages: (**a**) total aerobic counts, (**b**) lactobacilli, (**c**) enterobacteria, (**d**) pseudomonads, (**e**) staphylococci and (**f**) yeasts/moulds. Points at zero represent levels below the detection limit (<1.50 log cfu/g for enterobacteria and <2.50 log cfu/g for pseudomonads and staphylococci). Error bars indicate standard deviations (*n* = 3).

**Figure 3 foods-08-00219-f003:**
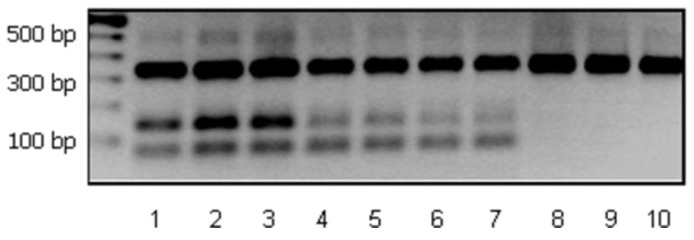
Molecular identification of *L. casei* ATCC 393 at levels ≥6 log cfu/g in probiotic dry-fermented sausages after 66 days of ripening applying strain-specific multiplex PCR assay after lactobacilli enumeration with microbiological analysis. Lane 1: Pure culture of *L. casei* ATCC 393 served as a positive control. Lane 2: Dry-fermented sausage containing 0.6% sugar and immobilized *L. casei* ATCC 393. Lane 3: Dry-fermented sausage containing 0.3% sugar and immobilized *L. casei* ATCC 393. Lane 4: Dry-fermented sausage containing no sugar and immobilized *L. casei* ATCC 393. Lane 5: Dry-fermented sausage containing 0.6% sugar and free *L. casei* ATCC 393. Lane 6: Dry-fermented sausage containing 0.3% sugar and free *L. casei* ATCC 393. Lane 7: Dry-fermented sausage containing no sugar and free *L. casei* ATCC 393. Lane 8: Dry-fermented sausage containing 0.6% sugar and no starter culture. Lane 9: Dry-fermented sausage containing 0.3% sugar and no starter culture. Lane 10: Dry-fermented sausage containing no sugar and no starter culture.

**Table 1 foods-08-00219-t001:** Effect of sugar addition and probiotic starter culture on shelf-life of dry-fermented sausages. Spoilage was defined as the point when 50% of the panellists rejected the sausage samples (spoilage was monitored up to 66 days).

Sugar	Starter Culture	Day of Spoilage
0.60%	Immobilized	No spoilage
	Free	No spoilage
	No culture	43
0.30%	Immobilized	49
	Free	47
	No culture	22
No sugar	Immobilized	43
	Free	43
	No culture	22

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
