# Peer review of "Effect of Sugar Content on Quality Characteristics and Shelf-Life of Probiotic Dry-Fermented Sausages Produced by Free or Immobilized Lactobacillus casei ATCC 393"

_foods, 2019, doi:10.3390/foods8060219_

Reviewer 1 Report

The manuscript investigated the effects of the sugar content (only two experimental points and the control) on the shelf life of sausages fermented with a Lactobacillus strain, either as such and immobilized on wheat grains (incidentally it is difficult to understood what the Author mean for immobilisation- see comments to the M&M section). The Authors  previously published other papers on the topic and the data presented here sounds like a hodgepodge information. Thus the work is poor and my personal impression is that the Authors  scratched the bottom of the barrel of their results. 

No interesting messages arose form the mist of statistical variations.

For these reasons the manuscript does not raise to the minimal standard of Foods journal.

Few notes.

Aim. it is not clear what the 'quality attributes' are for the Author.

Material & Methods are very poor of details (all paragraphs). Line 78: what 'noticeably' does mean? It is the control sample. How the spoilage was assessed?

The results are described in a trivial way (for example lines 113-118 are quite obvious)

line 149: enumeration stands for count?

Discussion: it is not clear the link between the claimed 'potential health benefits' and the determined 'quality parameters'

Author Response

Reviewer 1.

The manuscript investigated the effects of the sugar content (only two experimental points and the control) on the shelf life of sausages fermented with a Lactobacillus strain, either as such and immobilized on wheat grains (incidentally it is difficult to understood what the Author mean for immobilisation- see comments to the M&M section). The Authors previously published other papers on the topic and the data presented here sounds like a hodgepodge information. Thus the work is poor and my personal impression is that the Authors scratched the bottom of the barrel of their results. 

No interesting messages arose form the mist of statistical variations.

For these reasons the manuscript does not raise to the minimal standard of Foods journal.

Few notes.

Answer: Although production of probiotic dry-fermented sausages with immobilized Lactobacillus casei was studied in our previous studies (), the effect of the initial sugar concentration on the quality characteristics and the shelf-life of probiotic dry-fermented sausages has not been investigated before. Determination of the optimum initial sugar concentration is a key parameter for new product development, especially in additive-free products, aiming at facilitating the fermentation process and ensure safety.

Following the reviewer’s comment, additional details about cell immobilization have been included in the Materials & Methods section in the revised manuscript.

Aim. it is not clear what the 'quality attributes' are for the Author.

Answer: Following the reviewer’s comment “quality attributes” has been changed to “quality characteristics” in the title of the revised manuscript.

Material & Methods are very poor of details (all paragraphs). Line 78: what 'noticeably' does mean? It is the control sample. How the spoilage was assessed?

Answer: Additional details have been included in the Materials & Methods section. Following the reviewer’s comment “noticeably” has been deleted in the revised manuscript.

The results are described in a trivial way (for example lines 113-118 are quite obvious).

Answer: Although the results might be obvious by the graphs of Figure 1, we think that they should be described in brief in the text. However, if the reviewer insists, we are willing to delete this part.

line 149: enumeration stands for count?

Answer: Yes, the sentence has been modified in the revised manuscript.

Discussion: it is not clear the link between the claimed 'potential health benefits' and the determined 'quality parameters'.

Answer: Following the reviewer’s comment the sentence has been altered in the revised manuscript.

Reviewer 2 Report

The study is interesting and industry applicable. However, the manuscript needs to be improved much. Please see my comments below.

Line 15: 0, 0.30 and

Missing coma. Change to         0, 0.30, and

Line 20: enterobacteria, staphylococci and pseudomonads

Missing coma. Change to enterobacteria, staphylococci, and pseudomonads

Line 22: 6 logcfu/g    Missing space after log

log cfu/g

Line 38-39: viability, activity and functionality

Missing coma after activity. Correct this punctuation mistake throughout the manuscript.

Line 40: In this vein

Did you mean ‘vain’?

Line 40: deliver vehicles

Delivery vehicles?

Lines 52-56: Although a few previous research efforts for inserting functional cultures in dry-fermented sausages are available [16, 19-22], there are still many concerns associated with survival of the probiotic strains in adequate amounts during the hard conditions dominating at the manufacturing process [21,22]. Thus, commercial application of probiotic microorganisms in fermented sausages is not available yet.

Restructure this paragraph, especially the last sentence.

Lines 68-71: Section 2.1. Bacterial strain and cell immobilization

Briefly explain the steps in immobilization process.

Line 76: prior sausages preparation

Prior to sausages preparation

Lines 72-80: 2.2. Production of probiotic dry-fermented sausages

Explain the sausage making process. What was the composition and ingredients of each batch of sausage? How many batches of sausage? How did you apply probiotic treatments? How did you store the sausage and how long?

Overall, what was your experimental design? What is the experimental unit in the current study?

Please clarify these.

Line 80: collected at various intervals and subjected to chemical, microbiological and molecular analysis.

Explain the intervals.

What kind of molecular analysis?

Line 81: 2.3. Physicochemical analysis.

Briefly explain the steps included in these measurements.

Line 84: 2.4. Microbiological analysis

What kind of microbiological analysis? When did you do the microbiological analysis? What day of storage?

Briefly describe the method used.

Lines 89-100: 2.6. Preliminary sensory evaluation

What type of sensory evaluation was this? Descriptive? If yes, mention it.

Was there a training session involved to standardize the attribute intensities?

Did you use any reference materials for various sensory attributes?

Was this paper based or computer based? Sensory room conditions?

Scale 0-5. Was this continuous or discrete?

Lines 101-107: 2.7. Experimental design and statistical analysis

You have 2 factors in the study; sugar content and probiotic application.

How did you apply these treatments at various levels? Explain the design. Was this a split plot design?

Figure.1. It looks like there were significant interactions between sugar levels and probiotic inclusion, however, no interactions have been properly explained with corresponding P values in the results.

Clarify in the result section,

Were there any main effects?

Were there any interactions?

How can we interpret the significant differences in these figures since the error bars represent standard deviation? No superscripts given.

Table.1: How did you determine day of spoilage? What was the criteria?

Fig.2: The initial counts of all bacteria on d1 looks very high (> 6 log). Explain this

Author Response

Reviewer 2.

Line 15: 0, 0.30 and

Missing coma. Change to         0, 0.30, and

Line 20: enterobacteria, staphylococci and pseudomonads

Missing coma. Change to enterobacteria, staphylococci, and pseudomonads

Line 22: 6 logcfu/g    Missing space after log

log cfu/g

Line 38-39: viability, activity and functionality

Missing coma after activity. Correct this punctuation mistake throughout the manuscript.

Answer: All linguistic errors have been corrected in the revised manuscript.

Line 40: In this vein

Did you mean ‘vain’?

Answer: The phrase has been deleted in the revised manuscript.

Line 40: deliver vehicles

Delivery vehicles?

Answer: The phrase has been corrected in the revised manuscript, as suggested by the reviewer.

Lines 52-56: Although a few previous research efforts for inserting functional cultures in dry-fermented sausages are available [16, 19-22], there are still many concerns associated with survival of the probiotic strains in adequate amounts during the hard conditions dominating at the manufacturing process [21,22]. Thus, commercial application of probiotic microorganisms in fermented sausages is not available yet.

Restructure this paragraph, especially the last sentence.

Answer: Following the reviewer’s comment, this paragraph has been restructured.

Lines 68-71: Section 2.1. Bacterial strain and cell immobilization

Briefly explain the steps in immobilization process.

Answer: Following the reviewer’s suggestion, the immobilization process is briefly described in the revised manuscript.

Line 76: prior sausages preparation

Prior to sausages preparation

Answer: The phrase has been corrected in the revised manuscript, as suggested by the reviewer.

Lines 72-80: 2.2. Production of probiotic dry-fermented sausages

Explain the sausage making process. What was the composition and ingredients of each batch of sausage? How many batches of sausage? How did you apply probiotic treatments? How did you store the sausage and how long?

Overall, what was your experimental design? What is the experimental unit in the current study?

Please clarify these.

Answer: Following the reviewer’s suggestion additional information on the production of probiotic dry-fermented sausages is included in the revised manscript.

Line 80: collected at various intervals and subjected to chemical, microbiological and molecular analysis.

Explain the intervals.

What kind of molecular analysis?

Answer: Additional details on the intervals and the molecular analysis are included in the revised manuscript, as suggested by the reviewer.

Line 81: 2.3. Physicochemical analysis.

Briefly explain the steps included in these measurements.

Answer: Physicochemical analysis is briefly described in the revised manuscript, as suggested by the reviewer.

Line 84: 2.4. Microbiological analysis

What kind of microbiological analysis? When did you do the microbiological analysis? What day of storage?

Briefly describe the method used.

Answer: Microbiological analysis is briefly described in the revised manuscript, as suggested by the reviewer.

Lines 89-100: 2.6. Preliminary sensory evaluation

What type of sensory evaluation was this? Descriptive? If yes, mention it.

Was there a training session involved to standardize the attribute intensities?

Did you use any reference materials for various sensory attributes?

Was this paper based or computer based? Sensory room conditions?

Scale 0-5. Was this continuous or discrete?

Answer: Additional information of preliminary sensory evaluation is included in the revised manuscript, as suggested by the reviewer.

Lines 101-107: 2.7. Experimental design and statistical analysis

You have 2 factors in the study; sugar content and probiotic application.

How did you apply these treatments at various levels? Explain the design. Was this a split plot design?

Answer: Three-way ANOVA was applied for physicochemical parameters and microbiological analysis (sugar addition, the nature of the starter culture and the ripening time were the 3 factors), while 2-way ANOVA in the preliminary sensory evaluation (sugar addition and the nature of the starter culture were the 2 factors). Please specify if additional details are required.

Figure.1. It looks like there were significant interactions between sugar levels and probiotic inclusion, however, no interactions have been properly explained with corresponding P values in the results.

Clarify in the result section,

Were there any main effects?

Were there any interactions?

How can we interpret the significant differences in these figures since the error bars represent standard deviation? No superscripts given.

Answer: Significant interactions are described in the revised manuscript, as suggested by the reviewer.

Table.1: How did you determine day of spoilage? What was the criteria?

Answer: The criteria used for spoilage determination are described in the revised manuscript, as suggested by the reviewer.

Fig.2: The initial counts of all bacteria on d1 looks very high (> 6 log). Explain this

Answer: The initial counts of enterobacteria, pseudomonads, and staphylococci were relatively high probably due to poor sanitation conditions, but they were significantly reduced during ripening.

Round  2

Reviewer 1 Report

Almost all the comment raised by the reviewers have been taken into account.

The Authors made huge efforts to amend  the manuscript and the point-to-point answers to the reviewer's comments are convincing.

The text improved substantially and, I have no further comments.